**Data Availability Statement:** All relevant data are within the manuscript and its Supporting information files.

# Sample-to-answer, extraction-free, real-time RT-LAMP test for SARS-CoV-2 in nasopharyngeal, nasal, and saliva samples: Implications and use for surveillance testing

**Kathryn A. Kundrod**[1☯], **Mary E. Natoli**[1☯], **Megan M. Chang**[1☯], **Chelsey A. Smith**[1], **Sai Paul**[1], **Dereq Ogoe**[2], **Christopher Goh**[2], **Akshaya Santhanaraj**[2], **Anthony Price**[3], **Karen W. Eldin**[4], **Keyur P. Patel**[5], **Ellen Baker**[3], **Kathleen M. Schmeler**[3], **Rebecca Richards-Kortum**[1]*

1 Department of Bioengineering, Rice University, Houston, Texas, United States of America, 2 Rice 360˚ Institute of Global Health, Rice University, Houston, Texas, United States of America, 3 Department of Gynecologic Oncology and Reproductive Medicine, The University of Texas MD Anderson Cancer Center, Houston, Texas, United States of America, 4 McGovern Medical School, The University of Texas Health Science Center, Houston, Texas, United States of America, 5 Department of Hematopathology, The University of Texas MD Anderson Cancer Center, Houston, Texas, United States of America

☯ These authors contributed equally to this work.
* rkortum@rice.edu

## Abstract

The global COVID-19 pandemic has highlighted the need for rapid, accurate and accessible nucleic acid tests to enable timely identification of infected individuals. We optimized a sample-to-answer nucleic acid test for SARS-CoV-2 that provides results in <1 hour using inexpensive and readily available reagents. The test workflow includes a simple lysis and viral inactivation protocol followed by direct isothermal amplification of viral RNA using RT-LAMP. The assay was validated using two different instruments, a portable isothermal fluorimeter and a standard thermocycler. Results of the RT-LAMP assay were compared to traditional RT-qPCR for nasopharyngeal swabs, nasal swabs, and saliva collected from a cohort of patients hospitalized due to COVID-19. For all three sample types, positive agreement with RT-LAMP performed using the isothermal fluorimeter was 100% for samples with Ct <30 and 69–91% for samples with Ct <40. Following validation, the test was successfully scaled to test the saliva of up to 400 asymptomatic individuals per day as part of the campus surveillance program at Rice University. Successful development, validation, and scaling of this sample-to-answer, extraction-free real-time RT-LAMP test for SARS-CoV-2 adds a highly adaptable tool to efforts to control the COVID-19 pandemic, and can inform test development strategies for future infectious disease threats.

**Funding:** Funding for this work was provided from the American people by USAID through an IAVI research grant CCID 9204 under award AID-OAA-A16-00032 between IAVI and USAID, and through internal funding by Rice University. The funders had no role in study design, data collection and analysis, decision to publish, or preparation of the manuscript.

**Competing interests:** The authors have declared that no competing interests exist.

## Introduction

The global COVID-19 pandemic caused by the SARS-CoV-2 virus has demonstrated the need for highly accessible diagnostic and surveillance tests. Testing is a critical component of effective pandemic response and is used to identify outbreaks, inform contact tracing efforts, and contain viral spread. Diagnostic testing on individuals with suspected infections is necessary to provide appropriate clinical care and to maintain separation between individuals with and without confirmed SARS-CoV-2 infections, especially in hospital settings. In addition, SARS-CoV-2 transmission by individuals who are asymptomatic requires routine population-level surveillance for effective outbreak containment when rates of community transmission are high. Accordingly, SARS-CoV-2 testing capacity had to scale quickly to meet massive global demands.

The first diagnostic tests that received Emergency Use Authorization (EUA) from the United States Food and Drug Administration (U.S. FDA) were based on RT-qPCR, the standard-of-care method for detecting viral RNA. Traditionally, RT-qPCR testing requires sample collection into an appropriate medium, RNA extraction, and RT-qPCR reaction assembly prior to running the test on a thermocycler. Shortly after RT-qPCR testing became available across the country, supply shortages of sample collection kits, RNA extraction kits, RT-qPCR reagents, automated laboratory liquid handlers, and thermocyclers were common and unpredictable. Traditional RT-qPCR remains the standard-of-care method for SARS-CoV-2 diagnosis, but given the supply chain challenges and infrastructure required, alternative methods for surveillance were desirable.

A few key findings related to SARS-CoV-2 transmission cleared a path for alternatives to traditional RT-qPCR for surveillance testing. First, epidemiologic modeling indicated that test frequency mattered more than test sensitivity for effective outbreak prevention [1–3]. In addition, data showed "super-spreading", or transmission from presymptomatic or asymptomatic individuals with high viral loads, was a major driver of outbreaks [4–6]. Finally, test results from alternative specimen types like saliva were shown to correlate well with results from nasopharyngeal swabs and to be better indicators of infectivity [7–11]. Together, these findings informed the most critical goals of a SARS-CoV-2 surveillance program: frequent testing to detect asymptomatic individuals with high viral loads, and the flexibility to use alternative specimen types and methods that avoid the supply chain shortages associated with RT-qPCR.

Isothermal nucleic acid amplification technologies are a promising alternative to RT-qPCR due to their speed, reduced instrumentation requirements, and robustness to sample inhibitors. Of existing isothermal amplification methods, loop-mediated isothermal amplification (LAMP) is the most widely used globally [12]. Soon after SARS-CoV-2 sequences were published at the onset of the COVID-19 pandemic, scientists quickly developed reverse transcription LAMP (RT-LAMP) assays [13–17] and started global collaborations to develop and implement RT-LAMP for SARS-CoV-2 surveillance testing [18]. Given the high sensitivity, specificity, and concordance with RT-qPCR, RT-LAMP has been established as an effective tool to detect SARS-CoV-2, and has been incorporated into several tests that received EUA from the U.S. FDA [19–24]. However, existing RT-LAMP tests include complicated workflows that require RNA extraction [16, 25–28] or risk environmental cross-contamination [29], are costly [30], and/or have been validated with limited sample types and patient populations [17, 25, 27, 31–34], thus limiting their flexibility and adoption in at-scale surveillance testing.

Here, we present an RT-LAMP test for SARS-CoV-2 with a sample-to-answer workflow optimized for use with saliva, nasal, or nasopharyngeal swabs that can be performed with a standard thermocycler or portable isothermal fluorimeter. The test uses a simple lysis and inactivation protocol adapted from Rabe and Cepko [17] and provides results in <1 hour using inexpensive and readily available reagents. Test performance was compared to that of

RT-qPCR using nasopharyngeal swabs, nasal swabs and saliva obtained from patients hospitalized due to COVID-19. Following test validation, we piloted a saliva-based RT-LAMP testing program at Rice University and transitioned to a surveillance testing program. Between August 2020 and October 2021, up to 400 individuals were tested per day. The simple workflow, reliance on more readily available reagents, and effective detection of high viral load samples indicate that the optimized RT-LAMP assay presented here is an effective strategy for SARS-CoV-2 surveillance.

# Materials and methods

## Oligonucleotide primers and probes

The oligonucleotide primers for the SARS-CoV-2 RT-LAMP assay were the N2 and E1 primer sets designed by New England Biolabs, Inc. [35] and the Orf1a-HMSe (As1e) primer set designed by Rabe and Cepko [17], which target the N, E, and ORF1a genes. The ACTB primer set designed by New England Biolabs, Inc [35] to amplify the human beta actin gene was used as an extraction control. Primers were ordered from Integrated DNA Technologies (IDT) (Coralville, IA) either individually, or as a custom mix prepared by IDT using the same methods. FIP and BIP primers were HPLC-purified, and all remaining primers underwent standard desalting purification by IDT. Primers were resuspended in 1X TE buffer at a 1 mM concentration and combined to make a 25X mix consisting of 40 μM FIP and BIP, 5 μM F3 and B3, and 10 μM LF and LB in 1X TE buffer. 2019-nCOV CDC primer and probe mixes were purchased in EUA or RUO kits from IDT, and were used interchangeably in RT-qPCR.

## Target preparation and storage

Synthetic viral RNA from Twist Biosciences (San Francisco, CA, SKU 102019) was used as a positive control in RT-LAMP and RT-qPCR. Upon receipt, the concentration of control RNA obtained from Twist was determined by RT-qPCR using SARS-CoV-2 genomic RNA as a reference standard (ATCC®, Manassas, VA, VR-1991D™), and the RNA was stored in single-use 5 μL aliquots at -80˚C until use. Aliquots were diluted in nuclease-free water unless otherwise noted, kept on ice, and used within four hours of dilution.

NATtrol™ SARS-Related Coronavirus 2 (Zeptometrix, Buffalo, NY, NATSARS(COV2)-ST) was used as a pseudovirus control to assess the performance of candidate lysis buffers and the limit of detection of the RT-LAMP assay, and as a positive control for lysis during clinical sample testing. Pseudovirus was diluted to desired concentrations in nuclease-free water or an appropriate biological matrix before use.

Hs_RPP30 Positive Control plasmid from IDT was used as a positive control in RT-qPCR for the human RNase P assay, and human mixed gender genomic DNA (gDNA) (Promega, Madison, WI) was used as a positive control in the RT-LAMP beta actin assay. Positive controls were stored at 4˚C (NATtrol pseudovirus), or in single-use aliquots at ≤ -20˚C (Hs_RPP30 and gDNA), and diluted in nuclease-free water before use.

## RT-LAMP reactions

All RT-LAMP reagents were purchased from New England Biolabs (NEB, Ipswich, MA). Twenty-five μL RT-LAMP reactions to detect SARS-CoV-2 contained 12.5 μL of 2X E1700 Master Mix, 1 μL of 25X N2 primer mix, 1 μL of 25X E1 primer mix, and 0.5 μL of 50X fluorescent dye (NEB B1700). Unless otherwise noted, reactions also contained 1 μL of 25X As1e primer mix. RT-LAMP reactions to detect beta actin contained 12.5 μL of 2X E1700 Master Mix, 1 μL of 25X ACTB primer mix, and 0.5 μL of 50X fluorescent dye. All reactions contained

5 μL of sample unless otherwise noted, and reactions were supplemented with nuclease-free water to reach a final reaction volume of 25 μL. Reactions were run at 65˚C for 45 minutes in a Bio-Rad CFX96 thermocycler with heated lid set to 105˚C, and fluorescence measurements were taken every 10 seconds on the SYBR channel. Fluorescent drift baseline correction was applied to all data, and data were analyzed using the Bio-Rad CFX Maestro™ Software for CFX Real-Time PCR Instruments.

For RT-LAMP reactions run on the Axxin T8-ISO (T8), assay components described above were doubled to achieve 50 μL reaction volumes with two exceptions [36]: fluorescent dye was used at a 5X concentration instead of 50X [37], and 0.2 μL of Tte UvrD helicase (NEB) was included with each reaction [38, 39]. Reactions contained 10 μL of sample unless otherwise noted. Reactions were assembled in clear high-profile 8-tube strips, overlaid with 25 μL of molecular-grade mineral oil (Sigma Aldrich, St Louis, MO, 69794) and sealed with domed caps (Bio-Rad, Hercules, CA). Reactions were incubated at 65˚C on the Axxin T8 for 45 minutes, and fluorescence readings were taken every 10 seconds on the FAM channel, which was set to 7% PWM.

## RT-qPCR reactions

RT-qPCR reactions were set up using the Applied Biosystems TaqMan RNA-to-Ct 1-Step Kit (ThermoFisher, Waltham, MA) and 2019-nCov CDC EUA N1, N2, or RPP combined primer/probe mix (IDT). Each reaction contained 10 μL 2X RT-PCR Mix, 0.5 μL 40X RT Enzyme Mix, 1.5 μL combined primer/probe mix, 3 μL nuclease-free water, and 5 μL purified RNA template. Thermal cycling was performed according to manufacturer instructions, and amplification data were analyzed using the Bio-Rad CFX Maestro™ Software for CFX Real-Time PCR Instruments.

## Clean reaction setup

All amplification reactions were assembled and sealed prior to amplification in a dedicated pre-amplification room that was regularly decontaminated with bleach and RNaseAway™ (ThermoFisher) and had limited personnel access. Stringent training was implemented for laboratory personnel, separate filtered pipette tips were used for all materials, and gloves were changed frequently. Once reactions were run, reaction tubes were disposed of, without being opened, to prevent post-amplification contamination of future reactions.

## Lysis buffer optimization

**Lysis buffer preparation.** All lysis buffer reagents were of molecular or reagent grade. First, a concentrated TCEP/EDTA/NaOH buffer, adapted from Rabe and Cepko [17], was prepared as follows: 358 mg of TCEP-HCl (Millipore Sigma, Burlington, MA, 580567) were dissolved in 568 μL of nuclease-free water, and 1 mL of 0.5 M EDTA, pH = 8 (ThermoFisher AM9260G) was added. A 7 M GuHCl solution was prepared by adding 2,009 mg GuHCl (Promega H5381 or H5383, used interchangeably) to 1.5 mL nuclease-free water and adding additional nuclease-free water to a final volume of 3 mL. Next, 575 μL of 10 N NaOH (Fisher Scientific, Pittsburgh, PA, SS267) and 2,009 μL of 7 M GuHCl were added to the TCEP/EDTA solution, yielding 5 mL of 100X lysis buffer with 250 mM TCEP, 100 mM EDTA, 1.15 N NaOH, and 4 M GuHCl. Performance of the TCEP-based lysis buffer was evaluated at concentrations of 1X, 2X, 5X, and 10X, with and without the addition of GuHCl. For clinical sample collection, 100X lysis buffer with GuHCl was diluted 1:20 in nuclease-free water to yield 5X lysis buffer. All solutions were sterile-filtered through a 0.2-micron filter (Pall Life Sciences, Port Washington, NY, Acrodisc 4652), then stored at 4˚C for up to 7 days. pH was measured

for each batch, and a pH value less than 9.0 was considered adequate. If the buffer pH was greater than or equal to 9.0, all buffer reagents were replaced, and a new batch was prepared.

**Lysis efficiency evaluation.** Self-collected nasal swabs from asymptomatic donors were obtained with written informed consent under a protocol approved by the Rice University Institutional Review Board (2020–354). Nasal swabs were placed into 500 μL of nuclease-free water and vortexed for 20 seconds. Nasal swab eluate from at least five swabs were pooled to simulate a patient sample. Nasal swab eluate was spiked with Zeptometrix pseudovirus and combined with 100X lysis buffer and 7 M GuHCl to simulate 50 copies/μL and effective concentrations of 1X, 2X, 5X, or 10X lysis buffer components with a final effective concentration of either 0 or 200 mM GuHCl. In conditions without GuHCl, an appropriate volume of water was added such that nasal eluate volumes were consistent across conditions. Spiked samples in each candidate buffer were heat inactivated at 95˚C for 5 minutes, then column purified using the RNeasy Mini kit (Qiagen, Hilden, Germany) following the manufacturer's protocol for RNA cleanup with two modifications: PBS was substituted for Buffer RLT, and 15 second spin steps were increased to 1 minute. In the final step of the extraction protocol, RNA was eluted into 50 μL of nuclease-free water, then amplified using the 2019-nCov CDC EUA N1 RT-qPCR assay as described above. Ct values from baseline-subtracted, curve-fit data were determined by the Bio-Rad CFX Maestro software.

**RNase activity evaluation.** RNase activity of heat-inactivated spiked nasal samples was evaluated using the RNaseAlert QC Kit v2 (ThermoFisher) according to manufacturer instructions. Briefly, 10 μL of 10X RNaseAlert Lab Test Buffer, 10 μL of RNaseAlert substrate, 10 μL of sample, and 70 μL of nuclease-free water were combined in each well of an opaque-bottom black 96-well plate (Corning 3915). Negative (nuclease-free water) and positive (5 μL RNase A) controls were included. Samples were incubated in a BioTek Cytation 5 Cell Imaging Multi-Mode Reader at 37˚C, and real-time fluorescence data (Ex: 490 nm, Em: 520 nm) were collected at 5 minute intervals for a period of 1 hour. Data at the 1 hour timepoint were analyzed.

## Limit of detection

The limits of detection of the RT-LAMP assay were assessed by diluting Zeptometrix pseudo-virus in pooled negative nasopharyngeal matrix or pooled negative saliva. Pooled negative nasopharyngeal matrix was generated from 10 nasopharyngeal swabs that were confirmed negative by a clinical laboratory using the Cepheid Xpert System SARS-CoV-2 real-time PCR, or Applied Biosystems TaqPath COVID-19 real-time PCR tests using the MagNA Pure 24 Total NA Isolation Kit (Roche Molecular Systems, 07658036001). Each of these swabs was collected into 300 μL of 5X lysis buffer. Pooled saliva was generated by combining equal volumes of passive drool collected from 5 asymptomatic donors under a protocol approved by the Rice University Institutional Review Board (2020–354), and confirmed negative by RT-LAMP. Spiked samples prepared in nasopharyngeal matrix were diluted to concentrations of 1–10 copies/μL, vortexed for 20 seconds, then heat inactivated at 95˚C for 5 minutes. Spiked samples prepared in pooled saliva were combined 20:1 with 100X lysis buffer and 7 M GuHCl to achieve an effective 5X lysis buffer concentration, 200 mM GuHCl, and 2–20 copies/μL, then vortexed for 20 seconds and heat inactivated at 95˚C for 6 minutes. All samples were kept on ice before being input into RT-LAMP reactions for amplification on either the Bio-Rad CFX96 or the Axxin T8. Twenty replicates of each concentration were tested, and the percentage of replicates with positive amplification was calculated at each concentration. Probit analysis was then used to determine the limit of detection. Dose-effect functions were fit by XLSTAT for Microsoft Excel (Addinsoft, New York, NY) and the limit of detection was defined as the concentration for which there was a 95% detection probability.

## Participant samples

**Enrollment and sample collection from inpatients at Lyndon B. Johnson (LBJ) Hospital.** Nasopharyngeal, nasal, and saliva specimens were collected from hospitalized participants at Lyndon B. Johnson (LBJ) Hospital after obtaining written informed consent under a protocol reviewed and approved by The University of Texas MD Anderson Cancer Center, Harris Health and Rice University Institutional Review Boards (2020–0318). Participants were eligible if they (1) qualified for SARS-CoV-2 testing according to institutional criteria at the time of enrollment; (2) were willing and able to provide informed consent; (3) were able to perform protocol-required activities; and (4) were able to speak and read English or Spanish. All participants had confirmed SARS-CoV-2 infections by a hospital RT-qPCR test on a nasopharyngeal sample within 72 hours of enrollment in our study. In some cases, participants were enrolled before their hospital RT-qPCR test result was returned; if the result was negative, the participant was excluded from analysis.

Nasopharyngeal sampling was performed by a single provider using the methods described by Marty *et al.* [40] and remained consistent throughout the study. Briefly, swabs were held in place at the back of the nasopharynx for several seconds, then slowly removed while rotating. Nasal and saliva collection methods were optimized using samples from the first 41 participants enrolled in our study. Various nasal collection strategies were tested, which included self-collection or provider-collection, sampling one nostril or both nostrils, and sampling each nostril 5 or 10 times. Saliva collection strategies consisted of collection of drool directly into a pre-aliquoted volume of lysis buffer, either with swabs, sponges, or passive drool, and collecting saliva by passive drool into a sterile tube and subsequently metering the volume of saliva combined with lysis buffer.

Nasopharyngeal swabs were collected into 1.5 mL sterile Sarstedt screw-top tubes (Nümbrecht, Germany) containing 300 µL of 5X lysis buffer. Nasal samples were collected into 2 mL sterile Sarstedt screw-top tubes containing 500 µL of 5X lysis buffer. Due to the shape of the swabs, a different tube and slightly more volume was required to fully submerge the nasal swabs compared to the nasopharyngeal swabs. Following collection, the shaft of the collection swab was cut at the height of the tube, and the tube was closed. Saliva samples were collected by passive drool into empty sterile tubes, unless otherwise indicated. All samples were stored in coolers with ice packs until arrival at the testing lab, which was within four hours of collection.

**Processing of samples collected at LBJ Hospital.** Upon receipt, nasopharyngeal and nasal samples were vortexed for 20 seconds and heated in a heat block at 95˚C for 5 minutes. Saliva samples collected into lysis buffer were processed similarly, but heating was extended to 6 minutes. Raw saliva samples collected without lysis buffer were combined 20:1 with 100X lysis buffer containing GuHCl, vortexed for 20 seconds, and heated at 95˚C for 6 minutes. Samples were kept on ice following heat inactivation until testing. For each sample, three replicates were tested in RT-LAMP on the Bio-Rad CFX96, and one replicate was tested in RT-LAMP on the Axxin T8. All experiments in which patient samples were tested included a positive control of synthetic RNA prepared in 5X lysis buffer at 10–100 copies/µL, and a no-template control using water. After saliva processing conditions were optimized, a lysis control of Zeptometrix pseudovirus in pooled saliva (Innovative Research, Novi, Michigan) was also included.

To quantify Ct values of all samples collected at LBJ Hospital, RNA was extracted from 200 µL of each heat-inactivated clinical sample using the PureLink Viral RNA/DNA Mini Kit (Invitrogen, Carlsbad, CA) according to manufacturer's instructions. Carrier RNA was not included in the extraction and purification procedure. Purified RNA was eluted in 50 µL

nuclease-free water, then amplified in the 2019-nCov CDC EUA N1, N2, and RPP RT-qPCR assays with both no-template controls and positive SARS-CoV-2 (Twist RNA) and RPP plasmid controls (IDT). Ct values from baseline-subtracted, curve-fit data were determined by the Bio-Rad CFX Maestro software.

Following initial testing, all samples were stored at -80°C, and samples used for retesting underwent a maximum of 3 freeze-thaw cycles.

**Enrollment and sample collection from asymptomatic individuals undergoing surveillance testing.** Members of the Rice University community without symptoms of COVID-19 underwent weekly or twice-weekly surveillance testing per university guidelines beginning in August 2020. To evaluate the performance of the developed RT-LAMP test when implemented in a high-throughput format, members of the Rice University community were offered the opportunity to complete their testing requirement by enrolling in a research protocol that was reviewed and approved by the Rice University Institutional Review Board (2021–28). Eligible participants included Rice University students, staff, or faculty who were not exhibiting symptoms of COVID, had no history of a positive nucleic acid, antigen, or antibody test result for SARS-CoV-2, and were willing and able to provide informed consent.

Following initial evaluation of the high-throughput RT-LAMP test, the test was piloted and then offered routinely as one of several options available to Rice University individuals for required campus-wide surveillance testing, following guidelines for universities established by the Centers for Medicare and Medicaid Services [41]. For minors, authorization was provided by the minor, parent or guardian.

Participants visited a dedicated collection site, where staff assisted in completing written informed consent forms during the piloting phase, or written testing authorization forms during the surveillance phase. Participants then provided saliva samples by passive drool into a barcoded, sterile 5 mL tube (MTC Bio or Eppendorf) and placed their barcoded sample into a cooler with ice. Samples remained on ice for up to four hours before they were transported to a high-throughput RT-LAMP lab for testing.

## High-throughput RT-LAMP setup

The high-throughput lab was equipped with four Hamilton Microlab Prep instruments and four Bio-Rad CFX96 thermocyclers. Saliva samples were combined with lysis buffer and inactivated manually, using the same heating methods as described above. RT-LAMP master mix was prepared and manually plated into 96-well plates. Plates were placed onto Isofreeze cold blocks (Thomas Scientific, 1148D62) for the duration of assay setup. Inactivated samples were added to RT-LAMP master mix by a Hamilton Microlab Prep within 30 minutes of the time that the master mix was prepared. The Microlab Prep liquid handler was programmed to transfer 5 µL of inactivated saliva from each participant into a distinct master-mix containing well on a 96-well plate. The transfer step included mixing by pipetting during the transfer. The plate was sealed manually with an optically-clear MicroSeal B PCR plate sealing film (Bio-Rad MSB1001), spun in a plate quick-spin centrifuge for 30 seconds (Fisherbrand 14-955-300), and placed onto the Bio-Rad CFX96 for amplification and detection using the same settings previously described for RT-LAMP reactions.

One replicate was run per-person, and any suspected positives were re-run in triplicate. Samples were determined to be positive or negative using the algorithm described in the Data analysis section below. Participants with a positive saliva RT-LAMP test were scheduled for a confirmatory nasal RT-qPCR test from a contracted testing partner.

With every run of the SARS-CoV-2 assay, a subset of participant samples was also tested in the beta actin RT-LAMP assay as a plate and process control designed to help identify issues

like sample degradation during transport, insufficient lysis, or LAMP reagent quality deterioration. If greater than 20% of participant samples included in the beta actin assay failed to amplify, lysis buffer was discarded and re-prepared with fresh reagents.

Study data, including consent forms, participant information, and test results, were collected and managed using REDCap (Research Electronic Data Capture version 10.9.2) electronic data capture tools hosted at The University of Texas MD Anderson Cancer Center or Rice University [42, 43]. REDCap is a secure, web-based software platform designed to support data capture for research studies, providing 1) an intuitive interface for validated data capture; 2) audit trails for tracking data manipulation and export procedures; 3) automated export procedures for seamless data downloads to common statistical packages; and 4) procedures for data integration and interoperability with external sources.

The unused portion of all samples was stored at -80˚C, and samples used for retesting underwent a maximum of 3 freeze-thaw cycles [44].

## Data analysis

**Bio-Rad CFX96 RT-LAMP data.** For all RT-LAMP data acquired on Bio-Rad thermocyclers, fluorescent drift baseline correction was applied to baseline-subtracted, curve-fit data by the Bio-Rad CFX Maestro™ Software. In some cases, fluorescent drift baseline correction resulted in negative slopes. In these cases, Ct values from uncorrected fluorescence data were used in subsequent analysis. Reactions given a Ct value by the software prior to the end of the incubation period were considered positive. Samples tested in triplicate were considered positive if all three replicates were given a Ct value, or negative if none of the three replicates were given a Ct value. If one or two of three replicates were given a Ct value, the following algorithm was applied: (1) If any of the three replicates amplified before 30 minutes ($<$ 81 cycles), the sample was considered positive. (2) If at least one of the three replicates amplified between 30 and 45 minutes (81–121 cycles), both of the following conditions had to be met for the sample to be considered positive: (A) the melt curve derivative peak of at least one replicate was within 1˚C of the melt curve derivative peak of a positive control run on the same plate and (B) the melt curve derivative remained positive between 65–70˚C. (3) Otherwise, the sample was considered negative.

For high-throughput LAMP samples run with a single replicate, the sample was suspected positive if it amplified before 30 minutes, or if it amplified between 30 and 45 minutes and the melt curve derivative peak was within 1˚C of the melt curve derivative peak of a positive control run on the same plate and the melt curve derivative remained positive between 65–70˚C. Suspected positives were subsequently re-tested in RT-LAMP in triplicate, and the above algorithm was applied to determine positivity.

**Axxin T8 RT-LAMP data.** For all RT-LAMP data acquired on Axxin T8, baseline subtraction was applied to each reaction by averaging the raw fluorescence value between 2–5 minutes and subtracting the averaged value from all of the fluorescence data points from that reaction. Samples were considered positive if (1) the amplitude of the baseline-subtracted data reached $>$ 500 millivolts (mV) between 5 and 45 minutes, and (2) the maximum slope of the baseline-subtracted data between 0 and 45 minutes was greater than 100 mV/s. Samples were considered negative if both conditions were not met. The time to positivity was defined as the time at which the amplitude of the baseline-subtracted data reached $>$ 500 mV. Amplification curves in figures are shown with relative fluorescence units (RFU) for clarity, where 1 RFU = 1 mV as reported by the T8-ISO.

**Statistical analyses.** Data were initially analyzed using Microsoft Excel (version 16.48), then input into GraphPad Prism (version 9.1.0) for statistical tests and data visualization. 95%

confidence intervals were calculated with the modified Wald method using GraphPad QuickCalc.

## Results

### Sample-to-answer workflow

The developed test combines extraction-free sample lysis and inactivation with isothermal amplification using RT-LAMP for real-time detection of SARS-CoV-2, and is suitable for testing in near point-of-care settings (Fig 1A–1F). In a near point-of-care use case, the test requires sample collection and lysis tubes, prepared lysis buffer, a heat block, pipettes,

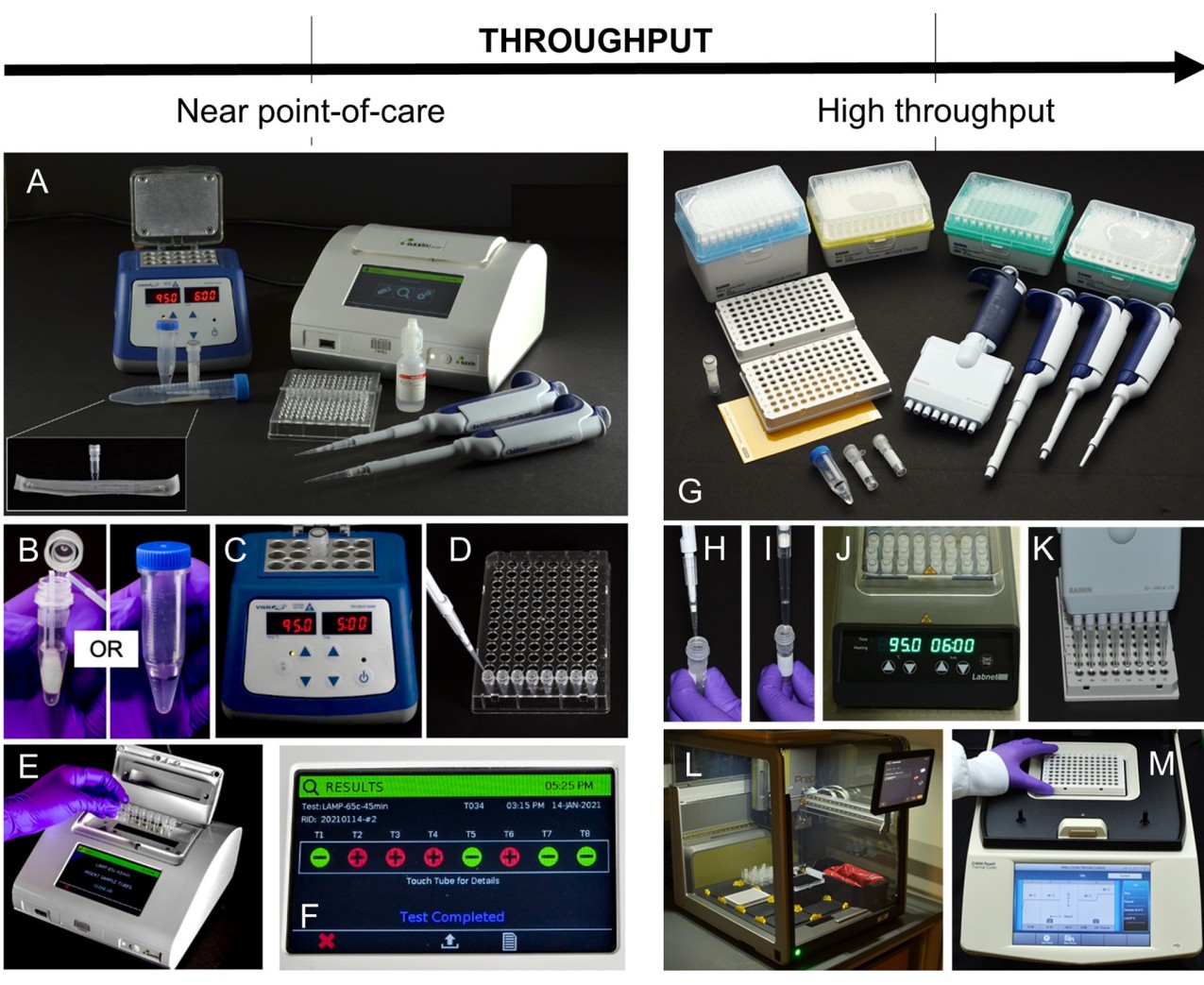

**Fig 1. Near point-of-care and high-throughput sample-to-answer workflows.** (A-F) Near point-of-care workflow. (A) Equipment and consumables needed to run the RT-LAMP assay, including saliva collection and lysis tubes, a heat block, pipettes, mineral oil, and a benchtop fluorimeter. A nasal swab and lysis tube (inset) can be substituted for saliva collection materials. To perform the test, (B) either a nasal swab or saliva sample is collected and combined with lysis buffer, then (C) the sample is heated at 95°C for 5 (nasal) or 6 (saliva) minutes. (D) The lysed sample is then added to RT-LAMP reagents and (E) reactions are sealed and placed in a benchtop fluorimeter and heated at 65°C for 45 minutes with real-time fluorescence monitoring. (F) Results are displayed after the full incubation period. (G-M) High-throughput saliva workflow. (G) Consumables and pipettes needed to run the RT-LAMP assay, excluding Hamilton robotics consumables. To perform the test, (H) lysis buffer and (I) saliva are combined and (J) heated at 95°C for 6 minutes. (K) RT-LAMP master mix is prepared and added to a 96-well plate. (L) Inactivated samples are added to RT-LAMP master mix by a Hamilton MicroLab Prep. The plate is sealed and (M) placed onto the Bio-Rad CFX96 for amplification and detection. Results are analyzed within the Bio-Rad Maestro software.

RT-LAMP reagents, and a real-time isothermal fluorimeter, such as the Axxin T8 (Fig 1A). Sample collection, inactivation, and amplification requires <5 user steps per sample, and results can be obtained in <1 hour. Moreover, the assay can be adapted to accommodate higher throughputs needed for population surveillance by incorporating automation and real-time thermocycling machines, such as the Bio-Rad CFX96 (Fig 1G–1M), while still maintaining rapid time to result. The higher throughput format of the developed RT-LAMP test was implemented in an asymptomatic surveillance population at Rice University.

## RT-LAMP primer set evaluation

First, a multiplexed RT-LAMP assay was validated using previously published primer sets N2, E1 and As1e targeting the N, E, and ORF1a genes, respectively. Recent literature suggests that the use of multiple primer sets in RT-LAMP increases sensitivity compared to single primer set reactions [45, 46], and that the use of N2/E1 and N2/E1/As1e can more than double the percentage of replicates that amplify at low viral loads [35]. Accordingly, we compared these multiplexed primer set combinations, and found that the two primer sets were comparably sensitive across all input concentrations tested. However, the combination of N2, E1, and As1e primer sets in RT-LAMP resulted in faster times to amplification, with a significant difference (p<0.001) seen at 50 input copies (S1 Fig). Thus, the triplex RT-LAMP assay was chosen due to its sensitivity, rapid time to amplification, and negligible increase in cost compared to the duplex RT-LAMP assay.

## Lysis buffer evaluation

To circumvent expensive and time-intensive nucleic acid extraction steps for a simple sample-to-answer workflow, we developed an extraction-free lysis and inactivation protocol. Recent work has demonstrated that TCEP/EDTA buffers, in combination with heat treatment at 95˚C, are effective at viral lysis and endogenous RNase inactivation, and render patient samples safe for downstream handling [17, 47, 48]. Further, GuHCl, another strong lysis reagent and RNase inactivation reagent [49, 50], has been shown to increase sensitivity and speed in RT-LAMP with our chosen primer sets [35]. Thus, we evaluated a TCEP/EDTA lysis buffer at a range of effective concentrations between 1X-10X, with and without GuHCl, combined with a heating step at 95˚C for 5 minutes. We found that increasing buffer concentration resulted in a statistically significant increase in lysis efficiency, with 5X concentration achieving the most complete lysis (Fig 2A). Increasing lysis buffer concentration also led to more complete RNase inactivation, with the addition of 200 mM GuHCl leading to significantly more RNase inactivation (Fig 2B; p<0.0001). We concluded that a 5X concentration of lysis buffer with 200 mM GuHCl was optimal due to its efficient lysis and thorough inactivation of RNases. We validated that the buffer was well-tolerated for direct addition into LAMP by comparing time-to-amplification for RNA prepared in 5X lysis buffer with 200 mM GuHCl to that of RNA in nuclease-free water, and found that times-to-amplification were comparable (S2 Fig).

Given the challenges of obtaining nasopharyngeal swabs, the supply chain limitations for appropriate nasal swabs, and the emergence of saliva as a sensitive alternate specimen type, we next sought to adapt the lysis and inactivation protocol for saliva samples. However, initial experiments resulted in up to a 25-fold increase in the limit of detection for RT-LAMP with saliva samples relative to nasal and nasopharyngeal samples. We hypothesized that this higher limit of detection was due to a combination of incomplete lysis and a more inhibitory sample matrix. To investigate, we tested a range of heat times between 5 and 10 minutes using pooled saliva spiked with Zeptometrix at a concentration of 100 copies/μL. We found that heating times did not have a significant impact on time to amplification, but longer heat times did

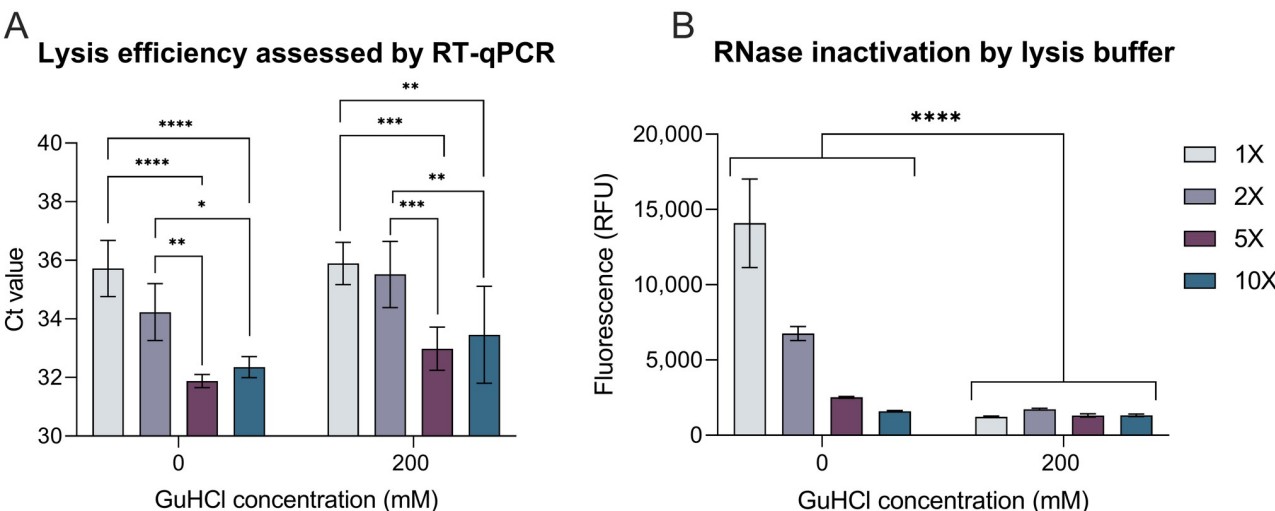

**Fig 2. Lysis buffer evaluation and optimization.** Four lysis buffer concentrations (1X, 2X, 5X, and 10X) with and without 200 mM GuHCl were evaluated. Candidate buffers were combined with negative nasal swab eluate, spiked with Zeptometrix pseudovirus and heated at 95° C for 5 minutes. (A) Average Ct values resulting from lysed samples amplified in the CDC N1 RT-qPCR assay following column purification (± standard deviation; n = 6 for each condition). Significance determined using a two-way ANOVA with post-hoc Tukey's test. (B) Average relative fluorescence values from lysed samples at the 1-hour timepoint of an RNase Alert assay (± standard deviation; n = 3 for each condition). Significance determined using two-way ANOVA. **** indicates $p < 0.0001$, ** indicates $p < 0.01$, * indicates $p < 0.05$.

result in less variable time to amplification (S3 Fig). The least variable amplification time was seen when saliva was heated for 6–7 minutes at 95°C. To minimize hands-on sample processing time, 6 minutes was chosen as the heat time with saliva samples.

## Limit of detection

Using the optimized lysis buffer and heating profiles, the limit of detection of the assay was found to be 20 virions per reaction on the Bio-Rad CFX 96 (Fig 3A) and 23 virions per reaction on the Axxin T8 (Fig 3B) in nasopharyngeal samples. In saliva, the limit of detection was 93 virions per reaction on the Bio-Rad CFX 96 (Fig 3C) and 116 virions per reaction on the Axxin T8 (Fig 3D). The limits of detection on the Bio-Rad CFX96 and the Axxin T8 were comparable, suggesting that modifications made in translating the assay to a point-of-care instrument did not impact sensitivity. Notably, the limit of detection is higher in saliva, which is a more complex sample matrix that has been shown to impact LAMP sensitivity [14, 31, 32, 51], probably due to the presence of mucins [52] and nucleases [53–55].

## Participant sample optimization and evaluation

A total of 94 participants from LBJ Hospital were enrolled between June 2020 and January 2021, from which nasopharyngeal, nasal, and/or saliva samples were collected and evaluated in both RT-qPCR and our RT-LAMP assay (Fig 4). 88 participants were confirmed to have SARS-CoV-2 infections by RT-qPCR on a nasopharyngeal swab obtained for clinical care purposes; of these, 10 participants did not provide nasopharyngeal swabs for research purposes, 2 did not provide nasal swabs, and 2 did not provide saliva samples, leading to a total of 78, 86, and 86 nasopharyngeal, nasal, and saliva samples collected for the study that were evaluated in RT-qPCR. A total of 35 nasal and saliva samples were used to optimize collection methods as described below, resulting in 41 participants with paired nasopharyngeal, nasal, and saliva samples collected under optimal conditions and assessed in RT-qPCR.

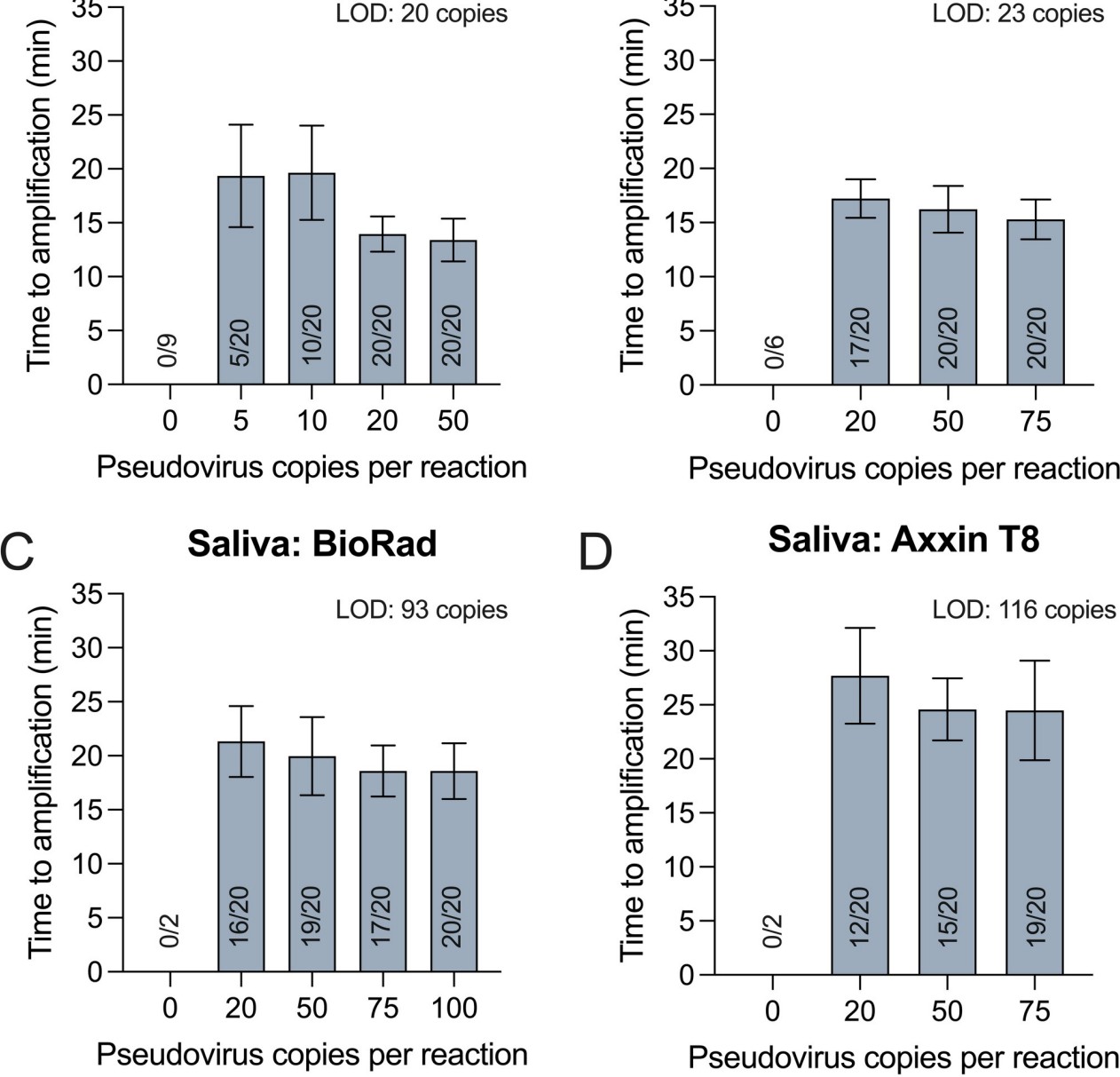

**Fig 3. Limit of detection in negative pooled matrix.** The limit of detection was assessed by spiking Zeptometrix pseudovirus into pooled nasopharyngeal matrix collected in lysis buffer that had previously tested negative for SARS-CoV-2 in RT-qPCR using the CDC assay, or negative pooled saliva combined with lysis buffer. Samples were heat-inactivated at 95° C for 5 minutes (A,B) or 6 minutes (C,D), and tested on either the Bio-Rad CFX96 or Axxin T8. The limit of detection at 95% probability was determined using probit analysis to be: (A) 20 copies per reaction on the Bio-Rad CFX96 and (B) 23 copies per reaction on the Axxin T8 for nasopharyngeal samples. The limit of detection in saliva was found to be: (C) 93 copies per reaction on the Bio-Rad CFX96 and (D) 116 copies per reaction on the Axxin T8.

**Nasal swab collection optimization.** Due to supply chain shortages of swabs, nasal samples were collected using a variety of swab types, and nasal swab collection methods were optimized throughout the course of our clinical sample collection period. We tested the effects of swab type (Puritan polyester swabs or Copan flocked swabs), swab collector (health-care

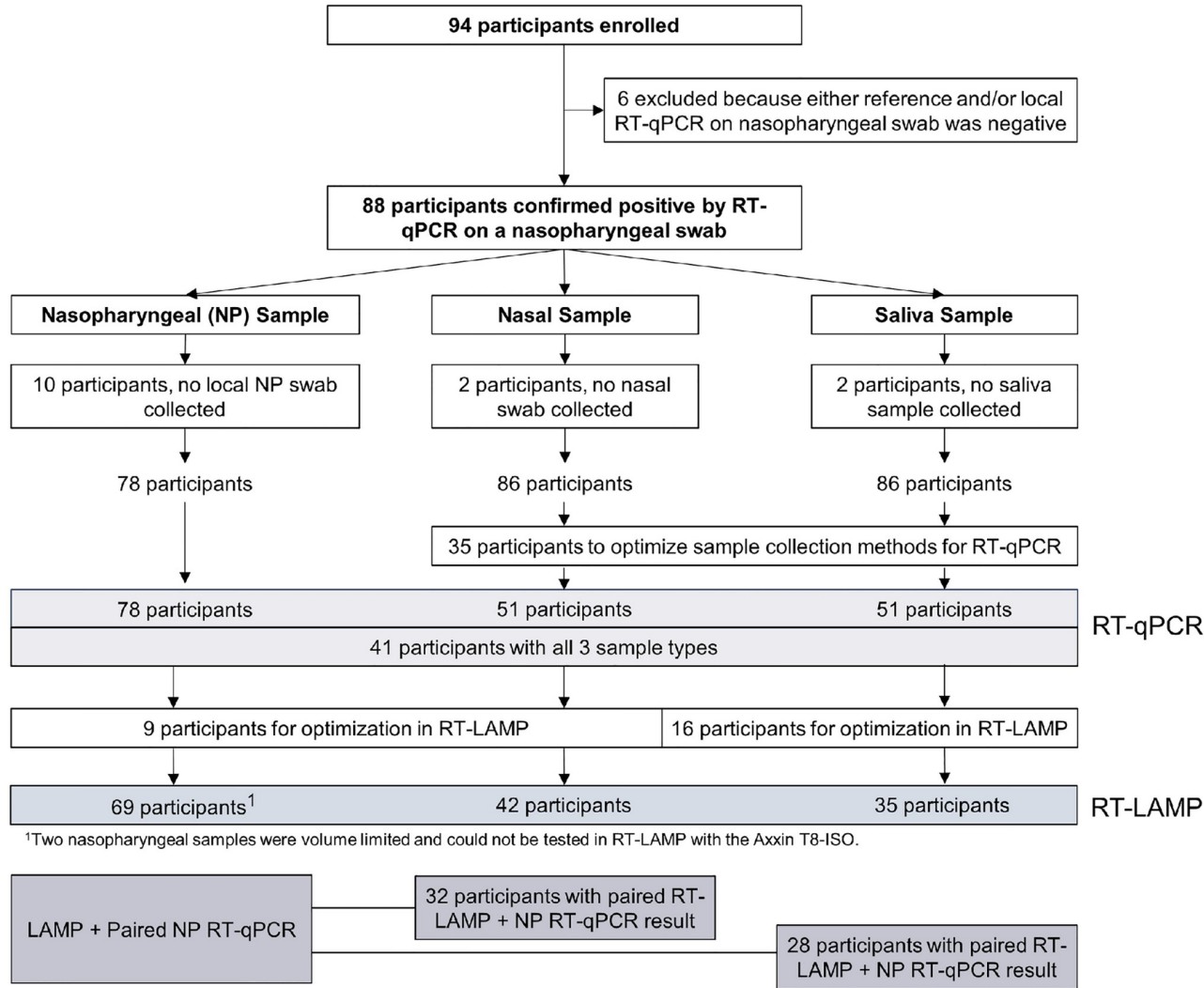

**Fig 4. Testing and analysis flowchart by sample type.** Flowchart shows the number of participants who were enrolled and confirmed to have SARS-CoV-2 infections, and the corresponding number of samples by type that were collected and analyzed by RT-qPCR and by RT-LAMP.

provider or patient self-collection), number of nostrils sampled (one or both), and number of rotations per nostril (5–10). We found that the swab type and swabbing method greatly impacted viral RNA detection in RT-qPCR when compared to the paired nasopharyngeal swab. Although nasal swabs required a slightly higher collection volume than nasopharyngeal swabs (500 μL vs. 300 μL), we anticipate that any differences in sensitivity attributed to this would be on the same order of magnitude and likely negligible. Previous reports have similarly detailed that specimen collection strategies impact test sensitivity [56]. Of the collection methods tested, the optimal strategy was found to be a provider-collected nasal sample using a Copan flocked swab that was rotated ten times in each nostril. This optimal collection strategy resulted in a positive RT-qPCR agreement with a paired nasopharyngeal swab of 71%, compared to an agreement of only 47% when using non-optimal swabs and collection methods (S4A Fig). The positive agreement for paired nasal and nasopharyngeal swabs undergoing RT-qPCR with the optimal sample collection method is comparable to that in other reports [57]. Samples collected under non-optimal conditions were excluded from subsequent analyses.

**RT-LAMP optimization with nasal swabs.** Initial results with optimized nasal swab collection methods indicated low concordance between RT-LAMP and RT-qPCR when nasal swabs had Ct>30. We hypothesized that RT-LAMP reactions were target-limited, and increasing the sample volume would increase the amount of viral RNA available for amplification. Therefore, we retested a subset of nasal samples (n = 4) that had initially discordant RT-LAMP and RT-qPCR results and/or RT-LAMP results in which fewer than three replicates amplified with sample volumes of 5 μL, 7 μL, and 9 μL (S4B Fig). Only one sample amplified in all three conditions, and sample volumes of 7 μL and 9 μL amplified more slowly than 5 μL. This suggested that increased sample volume was inhibitory to RT-LAMP, probably due to inhibition from lysis buffer components, particularly GuHCl. Thus, 5 μL was determined to be the optimal sample volume for nasal samples.

**Saliva collection optimization.** A variety of saliva collection and inactivation methods were used throughout our clinical study due to evolving biosafety considerations. Saliva collection and inactivation methods had notable effects on viral RNA detection in RT-qPCR and concordance with paired nasopharyngeal swabs. Initially, saliva was collected with swabs or sponges directly into tubes with 5X lysis buffer and heated before opening to ensure viral inactivation prior to sample handling. However, when RT-qPCR was performed on saliva samples collected in this manner, only 40% of these samples were positive by RT-qPCR despite having a positive nasopharyngeal swab (S5A Fig). This agreement between saliva and nasopharyngeal samples was much lower than previous reports [7, 11, 58]. We hypothesized that swabs and sponges did not collect sufficient cellular material, resulting in poor concordance.

We then began to collect saliva through passive drool into sterile tubes pre-loaded with a small volume of 100X lysis buffer; these tubes were vortexed and heated before sample handling. Although concordance with RT-qPCR increased to 55%, this was still considered to be an inadequate sample collection method. We attributed the low concordance to the variable ratio of saliva to lysis buffer, possibly resulting in incomplete RNase inactivation and/or viral lysis.

Finally, as laboratory biosafety guidelines for handling SARS-CoV-2 at our institution evolved, we were able to collect saliva into empty sterile tubes and combine saliva and lysis buffer in a controlled 20:1 ratio before heating. Under these conditions, concordance between RT-qPCR on paired nasopharyngeal samples increased to 78%, and thus this method was determined to be the best saliva collection and processing method. Saliva samples that were not collected under these conditions were excluded from subsequent analyses.

**RT-LAMP optimization with saliva.** Initial testing of saliva samples collected under the chosen best conditions showed poor RT-LAMP concordance with RT-qPCR results from the same sample. We observed that performance was worse with samples tested on the Axxin T8 than on the Bio-Rad CFX96, which we hypothesized to be due to the increased sample volume used with the Axxin T8, and accordingly, increased inhibitors in the reaction. Thus, we investigated whether reducing saliva sample volume could improve performance for RT-LAMP. A subset of five saliva samples with initially discordant results between RT-LAMP and RT-qPCR were retested at sample volumes of 2.5 μL, 3.75 μL, and 5 μL (S5B Fig). 5 μL volumes of saliva input into RT-LAMP resulted in amplification of only two of five samples, indicating that a 5 μL sample input may be inhibitory to the reaction. Similarly, only two of five samples amplified at input volumes of 2.5 μL, likely due to insufficient viral input. When tested at a sample volume of 3.75 μL, four of five samples had at least one positive replicate. Thus, 3.75 μL sample volumes were deemed to be optimal at maximizing viral input while minimizing inhibitors in the RT-LAMP reaction.

All saliva samples with sufficient remaining volume were tested with 3.75 μL sample input volumes on the Bio-Rad CFX96 and with 7.5 μL sample input volumes in 50 μL reactions on

the Axxin T8. RT-LAMP results were then compared to initial results obtained with higher sample volumes (S6 Fig). For the Bio-Rad CFX96, concordance with RT-qPCR was similar at both sample input volumes, but time to amplification was on average 6.6 min faster at the lower sample volume (S6A Fig). Decreasing sample volume in RT-LAMP reactions on the Axxin T8 increased concordance with RT-qPCR, and reduced average time to amplification by 5.9 min. Therefore, 3.75 μL and 7.5 μL sample volumes were selected as the best volumes for subsequent RT-LAMP performance evaluation on the on the Bio-Rad CFX96 and the Axxin T8, respectively.

Importantly, the re-optimization undertaken here highlights the value of testing real patient samples rather than contrived samples, especially with complex sample matrices such as saliva, as contrived samples cannot capture the person-to-person variability nor disease-induced changes in composition [55], pH [48], or viscosity [59] that may occur.

## Heterogenous Ct distribution among participant samples

The distributions of Ct values by sample type were plotted for 41 participants from whom all three sample types were collected under optimal conditions (Fig 5A–5C). The peak of the distribution of Ct values was lowest for nasopharyngeal swabs and highest for nasal swabs. The highest variability in Ct values was observed for saliva samples.

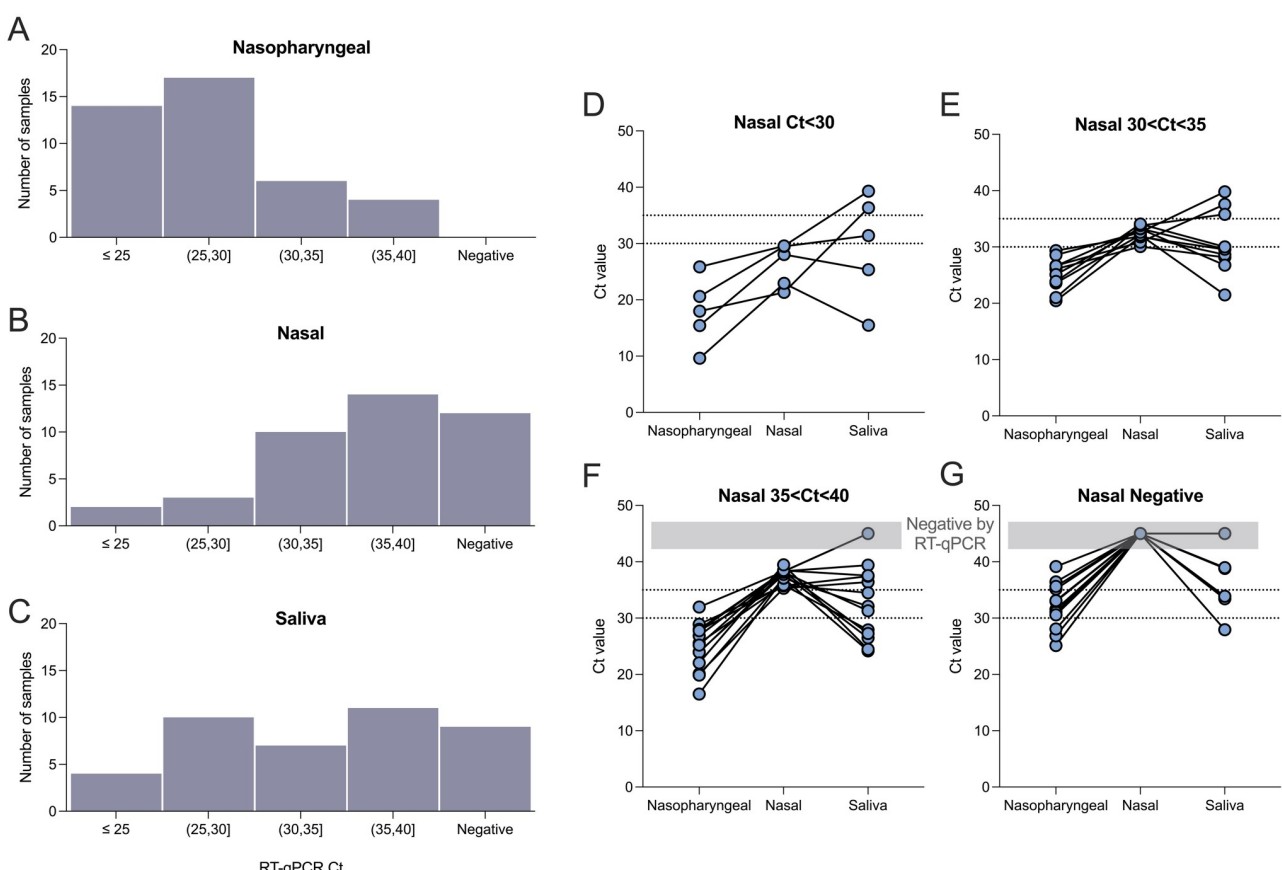

**Fig 5. Ct distribution by sample type.** The frequency distribution of Ct values obtained by RT-qPCR for (A) nasopharyngeal swabs, (B) nasal swabs, and (C) saliva samples from patients who provided all three sample types (n = 41). The Ct values of corresponding nasopharyngeal, nasal, and saliva samples from the same patients are plotted and stratified by (D) nasal swab Ct<30, (E) nasal swab 30<Ct<35 (F) nasal swab Ct>35 and (G) negative nasal swab.

To investigate the relationship between sample type and viral load, participant samples were divided into four groups based on the Ct value measured for the nasal swab, and the corresponding nasopharyngeal, nasal, and saliva Ct values were plotted (Fig 5D–5G). Nasal swabs from the same patient consistently had higher Ct values than the corresponding nasopharyngeal swab, whereas the Ct value of the corresponding saliva sample was variable. These trends in viral load distribution among the sample types likely reflect that the patients tested were late in their course of infection [60], and support findings that viral RNA is persistently detected in nasopharyngeal swabs longer than saliva [7, 57, 60, 61] or nasal samples [57, 60].

## Optimized RT-LAMP test performance on participant samples

A total of 69 nasopharyngeal swabs, 42 nasal swabs, and 35 saliva samples were tested using the optimized RT-LAMP assay. Inactivated lysate was directly input into RT-LAMP for evaluation on both the Bio-Rad CFX96 and the Axxin T8. Samples were also purified and assessed in RT-qPCR, and Ct values obtained from the N1 and N2 assay were averaged for each sample. RT-LAMP results were then compared to RT-qPCR to assess positive and negative agreement at varying Ct thresholds.

For nasopharyngeal swabs, RT-LAMP on the Axxin T8 showed overall positive agreement of 91% (95% confidence interval [CI]: 81–96%) with RT-qPCR. Positive agreement with RT-qPCR was 95% (95% CI: 86–99%) for nasopharyngeal samples with Ct<35, and 100% (95% CI: 90–100%) for samples with Ct<30. Comparable results were obtained for RT-LAMP performed on the Bio-Rad CFX96 (Fig 6A–6C).

For nasal swabs, agreement between RT-LAMP and RT-qPCR was similarly high for samples with Ct<30 (100%, 95% CI: 56–100%) and Ct<35 (100%, 95% CI: 72–100%) on the Axxin T8 (Fig 6D–6F). Overall, agreement was 69% for samples tested on the Axxin T8 (95% CI: 50–84%), due in part to the high number of nasal samples with Ct>35 in the population tested. Similar positive agreement results were obtained on the Bio-Rad CFX96, and negative agreement on both instruments was 94% (95% CI: 70–100%).

For saliva, agreement between RT-LAMP and RT-qPCR was 100% (95% CI: 76–100%) for samples with Ct<30, 86% (95% CI: 65–96%) for samples with Ct<35, and 69% (95% CI: 51–83%) overall for samples tested using the Axxin T8. Comparable results were obtained on the Bio-Rad CFX96 (Fig 6G–6I). Concordance between RT-LAMP and RT-qPCR was slightly lower for saliva than for nasal swabs, which we attribute to the higher limit of detection for saliva than nasal and nasopharyngeal samples noted earlier, and the variability of saliva as a clinical matrix.

Overall, RT-LAMP results in this hospitalized cohort of participants showed good concordance with RT-qPCR from the same sample; however, although nasopharyngeal swab collection is impractical for surveillance testing, it is considered the most stringent reference standard. For this reason, we sought to evaluate differences in sensitivity of nasal RT-qPCR, nasal RT-LAMP, and saliva RT-LAMP. We compared performance of these three testing methods within the same cohort of hospitalized patients, using nasopharyngeal swab RT-qPCR as the reference standard and stratifying results by the reference Ct value. The sensitivity of nasal and saliva RT-LAMP and nasal RT-qPCR exceeded 90% for participants whose nasopharyngeal swab had a Ct <25 (S7 Fig). For participants with a nasopharyngeal swab Ct<30, the sensitivity of RT-LAMP for both saliva and nasal swabs was approximately 60%, and approximately 50% at Ct <35. In this hospitalized cohort of patients, the overall sensitivity of RT-LAMP was 54% (95% CI: 36–70%) for saliva samples and 47% (95% CI: 31–64%) for nasal swabs, both less than the corresponding 71% sensitivity of nasal RT-qPCR (95% CI: 55–83%).

Considering public health recommendations emphasizing frequency of testing over test sensitivity [1–3], we implemented our developed RT-LAMP saliva workflow as part of Rice's

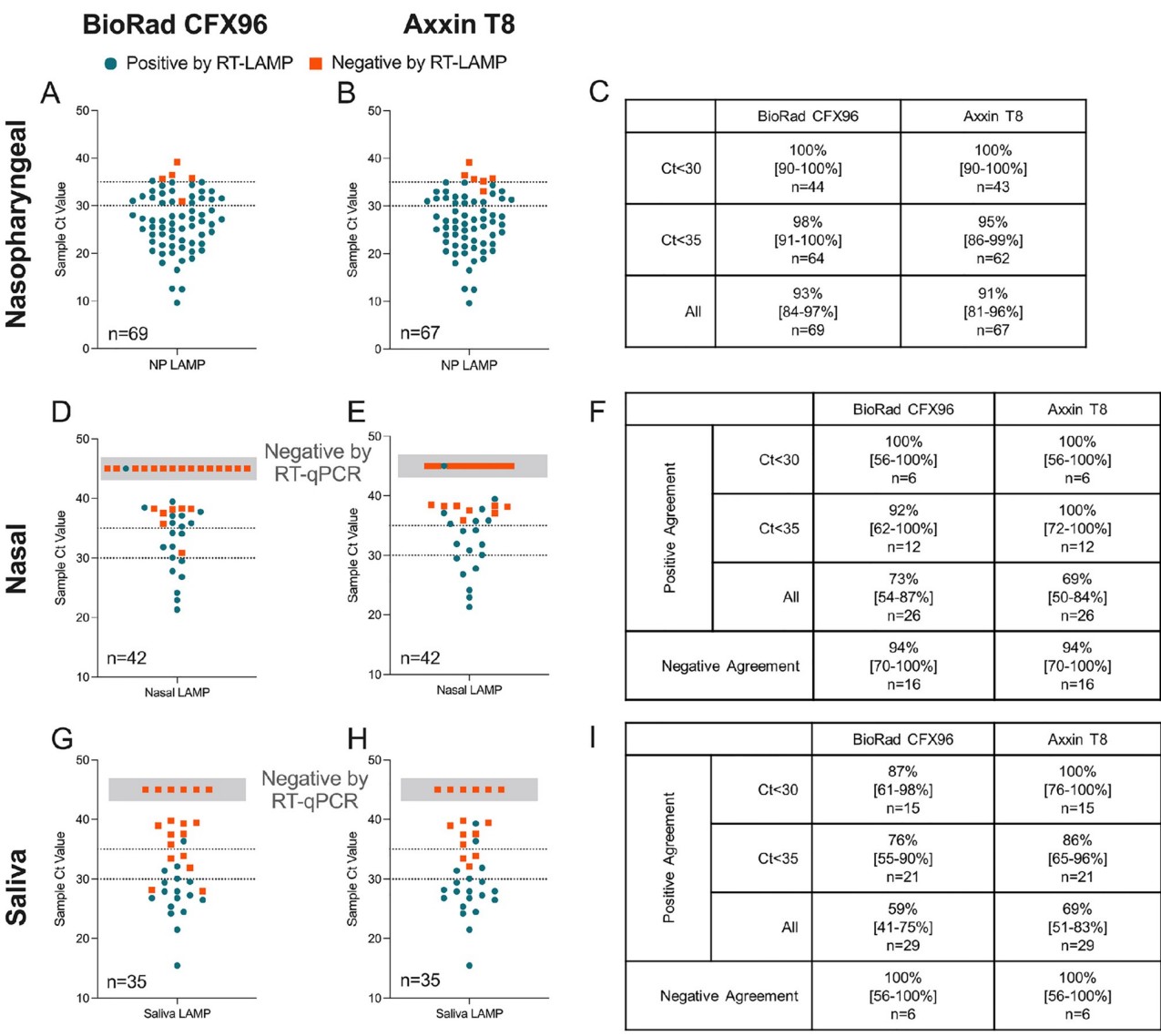

**Fig 6. RT-LAMP performance on clinical samples.** (A,B) Summary of RT-LAMP results for nasopharyngeal samples, stratified by the Ct value obtained in RT-qPCR. Nasopharyngeal samples were tested on (A) the Bio-Rad CFX96 and (B) the Axxin T8. (C) Positive agreement and 95% confidence intervals with RT-qPCR at varying Ct thresholds. (D,E) Summary of RT-LAMP results for nasal samples compared to the nasal sample Ct value obtained in RT-qPCR, when tested on (D) the Bio-Rad CFX96 and (E) the Axxin T8. (F) Positive and negative agreement with RT-qPCR and 95% confidence intervals for nasal samples. (G,H) Summary of RT-LAMP results for saliva samples compared to the saliva sample Ct value obtained in RT-qPCR, when tested on (G) the Bio-Rad CFX96 and (H) the Axxin T8. (I) Positive and negative agreement with RT-qPCR and 95% confidence intervals for saliva samples.

surveillance strategy. Saliva was chosen as the sample type for this application, due to comparable detection of SARS-CoV-2 in saliva and nasal swabs for samples with low Ct values (Fig 6), and practical advantages of saliva including circumventing nasal swab shortages and reliability of self-collected saliva.

## RT-LAMP as a surveillance test

As of December 16, 2021, a total of 24,927 participants at Rice University were tested in the high-throughput RT-LAMP assay. Thirty-seven people had presumptive positive test results,

and the remainder of the participants tested presumptively negative. Individuals who tested positive by saliva RT-LAMP, and five percent of individuals who tested negative by saliva RT-LAMP between September 2020 and January 2021, were directed to undergo a follow-up confirmatory nasal RT-qPCR within 24–48 hours. Data were available for nine of the 37 confirmatory tests, which were performed by one of Rice University's contracted COVID-19 testing providers on a self-collected nasal sample. Eight received a positive confirmatory test result and one received a negative confirmatory test result. The research team did not have access to laboratory results for the other 22 positive confirmatory tests which were performed by non-Rice affiliated community providers. All negative validation tests (n = 116) were confirmed negative.

Presumptive positive test results increased dramatically upon the arrival of the delta variant to Houston (S8 Fig), which occurred roughly between July 1 and November 1, 2021. The overall positivity rate prior to July 1 was 0.08%; the positivity rate between July 1 and November 1, 2021 was 0.20%. Additional changes over this period of time that might affect positivity rate include changes to campus-wide policies and testing intervals and the number of people on campus; however the trend seen in LAMP testing presumptive positivity was used in part to inform new guidance on masks and gathering size.

Ct values obtained from these nine confirmatory nasal RT-qPCR tests were summarized (Fig 7A). The average Ct value for nasal samples from these eight individuals was 21.1 ± 4.4, with a range between 16.7 and 28.2. This is slightly lower than the average (27.1 ± 7.2) and range (12.9–38.0) of Ct values for participants who received positive nasal RT-qPCR tests in Rice University's routine surveillance program (Fig 7A). Fig 7B compares the frequency distribution of Ct values for nasal swabs obtained in Rice University's routine surveillance program to that of the hospitalized cohort of patients enrolled in this study. Samples with low Ct values were more commonly found in the asymptomatic surveillance population than the hospitalized population, confirming other findings [65]. Over half of participants in the surveillance

A

|  | Confirmatory Nasal RT-qPCR (n=8) | Surveillance Nasal RT-qPCR (n=136) | Nasal RT-qPCR for hospitalized patients (n=35) |
|---|---|---|---|
| Mean | 21.11 | 27.05 | 33.17 |
| Standard Deviation | 4.39 | 7.24 | 4.74 |
| Range | 16.70-28.19 | 12.92-37.99 | 21.32-39.45 |

B

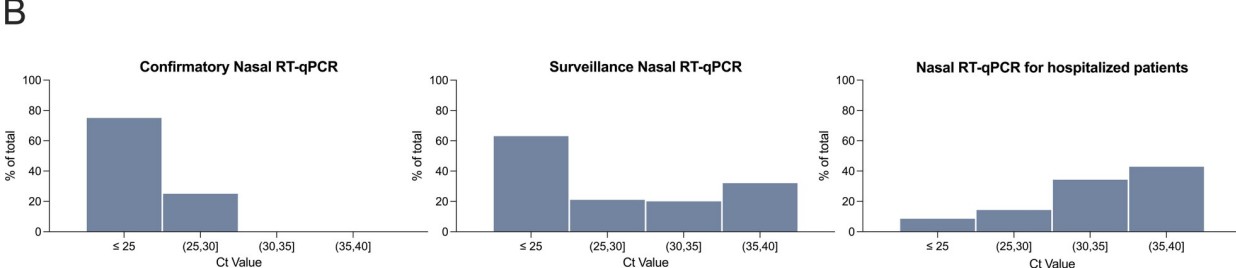

**Fig 7. Nasal Ct distributions in surveillance population and hospitalized patient population.** (A) Mean, standard deviation, and range of Ct values and (B) frequency distributions of Ct values for confirmatory nasal RT-qPCR tests after an individual tested positive with LAMP (n = 8), for individuals who tested positive as part of Rice University's surveillance testing program with the testing provider who conducted confirmatory RT-qPCR tests (n = 136), and for all hospitalized patients who tested positive by nasal RT-qPCR (n = 35).

population had Ct values $\leq 25$, whereas only $<10\%$ of hospitalized participants had Ct values $\leq 25$. Recent literature suggests nasopharyngeal swab Ct values above thresholds ranging from 24 to 35 are no longer infectious [62–65], highlighting the importance of a surveillance test that accurately detects samples with low Ct values.

## Discussion

We have described a sensitive and specific RT-LAMP test to detect SARS-CoV-2 RNA from unextracted swab and saliva samples. The inactivation protocol employed adds to the growing body of literature of TCEP/EDTA and heat-based strategies for viral lysis and endogenous RNase inactivation, and further demonstrates the compatibility of this method with multiple sample types.

The use of three unique primer sets in the RT-LAMP assay allows the test to target distinct regions of the SARS-CoV-2 viral genome, decreasing the likelihood that viral mutations would prevent amplification. The triplex assay achieves a low limit of detection of 20–23 copies per reaction, approaching that of RT-qPCR, for nasopharyngeal and nasal swab eluates. For saliva samples, the limit of detection is slightly higher, though still clinically relevant, at 93–116 copies per reaction. Our test falls within reported limits of detection in the literature for RT-LAMP SARS-CoV-2 tests, which range from 5–500 copies per reaction [16, 17, 25–28, 31, 32, 35, 48, 66–68], and is comparable to commercially available point-of-care tests [69–71]. To the best of our knowledge, RT-LAMP assays developed thus far for SARS-CoV-2 detection in clinical samples have largely relied on two primer sets targeting the viral genome for increased sensitivity, either in separate tubes [28, 29, 34, 48, 68] or in a duplex reaction [25, 29, 32, 33, 46, 72]. Few reports have explored triplex reactions [26, 35, 45].

In this study, we demonstrate that the limits of detection of the assay on the Bio-Rad CFX96 and the Axxin T8 are comparable, suggesting that modifications made in translating the assay to a point-of-care instrument did not impact sensitivity. The Axxin T8 requires a modest up-front equipment cost around $6,500 USD, can be powered by batteries, and is reusable, eliminating much of the plastic waste that will be generated by single-use at-home or point-of-care SARS-CoV-2 tests. Furthermore, reaction vessels remain closed, and amplicons never need to be manipulated for analysis, reducing the likelihood of environmental cross-contamination [73]. Successful implementation and validation of the RT-LAMP assay on the Axxin T8 has overcome the challenges of adapting assays to point-of-care instruments [36] and shows the potential of the use of this workflow near the point-of-care. FDA Emergency Use Authorization and widespread use of the Lucira COVID-19 All-In-One Test Kit at the time of publication illustrates the potential for similar technologies to be translated for true point-of-care use. To facilitate use in settings with limited laboratory infrastructure or trained personnel, RT-LAMP reaction mixtures would need to be lyophilized [74–76] to eliminate the manual reaction assembly that is currently required. Additionally, the saliva-specific workflow involves handling potentially infectious material, as the lysis and inactivation protocol requires users to combine saliva with lysis buffer before heating. The development of a collection and lysis module that meters saliva into lysis buffer could address this limitation.

The developed RT-LAMP test showed positive agreements of 87–100% for Ct $<30$ and 76–100% for Ct $<35$ compared with the hospital gold standard RT-qPCR test across all sample types and instruments, highlighting the utility of the test in the reported range of Ct values that are correlated with infectivity ($<24$–35) [62–65]. Recognizing that viral RNA persists much longer in nasopharyngeal specimens than in saliva and nasal swabs [7, 57, 60, 61], potentially affecting the Ct distributions from hospitalized patients who were tested in our study toward the end of their infection, we also analyzed the Ct values of positive samples identified through

our asymptomatic testing program. We found that nasal swab Ct values from the asymptomatic individuals trended lower than those measured from hospitalized patients, likely due to higher viral load occurring earlier in the course of infection [57, 63, 77–79]. As such, our results suggest that nasal or saliva samples likely work equally well in asymptomatic surveillance testing, when catching low-Ct value samples is the most important consideration. Nasopharyngeal swabs were a more sensitive sample type overall in our study, especially when people may have been late in the course of their infection, but the increased resources and personnel required to obtain nasopharyngeal swabs are not practical for asymptomatic surveillance. Our results add to other recent literature suggesting that sample type should be chosen carefully based on target population and testing goals.

Finally, the developed saliva RT-LAMP test was successfully scaled for surveillance testing of up to 400 members of the Rice University campus community per day. Scaling at Rice University involved sourcing dedicated equipment for testing, onboarding a team of laboratory technicians and student staff, and establishing sample collection and handling protocols. Sourcing thermocyclers and automated liquid handlers at the height of the pandemic was difficult and involved long lead times; these processes should be started as soon as the decision to perform in-house testing is made. Laboratory technicians relied on one to two liquid handlers and thermocyclers at a time to support testing of up to 400 people per day. Notably, the ability to rapidly optimize and validate the test in samples from hospitalized patients was essential to select a sample type and design the collection workflow. These lessons learned could be helpful in rapidly replicating surveillance testing for other emerging infectious diseases. Considering estimated equipment, reagent, and consumable costs, as well as personnel time, number of user steps, and limit of detection, the developed RT-LAMP test has a number of advantages over RT-qPCR tests, including a lower cost (S1 Table).

We have described a sample-to-answer test that involves a simple heat step for lysis and viral inactivation, transfer of the sample to amplification reagents, and incubation at a single temperature for nucleic acid amplification and real-time detection. We demonstrate two workflows, one that uses a portable isothermal fluorimeter and can be performed near the point-of-care to test tens of samples per day and a high-throughput format that uses a standard thermocycler to test hundreds of samples per day. The test is compatible with multiple sample matrices, including nasopharyngeal, nasal, and saliva samples. Compared with traditional RT-qPCR-based testing methods, this RT-LAMP test has the potential to circumvent supply chain bottlenecks for commonly used reagents and adds a highly adaptable method for surveillance testing for the COVID-19 pandemic.

## Supporting information

**S1 Fig. Multiplexed primer set selection.** Synthetic SARS-CoV-2 RNA was prepared in nuclease-free water and amplified in RT-LAMP on the Bio-Rad CFX96 using multiplexed primer sets N2/E1 or N2/E1/As1e. (A) Average time to amplification by number of input copies and primer set. The number of replicates that amplified out of 6 is indicated on each bar. Error bars indicate standard deviation of replicates that amplified. (B) Real-time amplification curves for 5, 10, 20, and 50 input copies of RNA per reaction. ***indicates p≤.001; significance determined using an unpaired two-tailed t-test.
(TIF)

**S2 Fig. Lysis buffer tolerance in RT-LAMP.** Synthetic SARS-CoV-2 RNA was prepared in either nuclease-free water or 5X lysis buffer with 200 mM GuHCl at a concentration of 10 copies/μL (50 copies per reaction) and amplified in RT-LAMP on the Bio-Rad CFX96. Average time to amplification ± standard deviation is shown (n = 3 for each condition). Time to

amplification is not significantly different between the two conditions (p≥0.6; significance determined using unpaired two-tailed t-test).
(TIF)

**S3 Fig. Saliva heat inactivation time.** Negative pooled saliva was spiked with Zeptometrix pseudovirus to achieve a concentration of 100 copies/μL, then combined with lysis buffer and heat inactivated at 95˚C for 5–10 minutes. The resulting lysate was amplified in RT-LAMP on the Bio-Rad CFX96. Average time to amplification was not statistically different between the heat inactivation time conditions (n = 6 per condition; p = 0.72 determined by one-way ANOVA) but variability of time to amplification was reduced at 6 and 7 minutes of heating.
(TIF)

**S4 Fig. Nasal optimization.** (A) Collection method optimization. Comparison of RT-qPCR results on nasal samples collected with various swabs and swabbing strategies. Nasal RT-qPCR results are stratified by the Ct value of the paired nasopharyngeal swab. (B) Ct (RT-qPCR) or average time to amplification in RT-LAMP (5 μL, 7 μL, 9 μL, indicating sample volume; n = 3 for each condition) of a subset of nasal samples that were initially discordant with RT-qPCR or inconclusive in RT-LAMP with 5 μL samples. Data shown by sample number in order of increasing Ct values; # indicates conditions where only 1 of 3 replicates amplified, ## indicates conditions where 2 of 3 replicates amplified.
(TIF)

**S5 Fig. Saliva optimization.** (A) Collection method optimization. Comparison of RT-qPCR results on saliva samples collected with various methods, including by swab or sponge, passive drool directly into buffer resulting in variable saliva to buffer ratios, or metering buffer into drool at a controlled ratio. Saliva RT-qPCR results are stratified by the Ct value of the paired nasopharyngeal swab. (B) Ct (RT-qPCR) or average time to amplification in RT-LAMP (2.5 μL, 3.75 μL, 5 μL, indicating sample volume; n = 3 for each condition) of a subset of saliva samples that were initially discordant with RT-qPCR with 5 μL samples. Data shown by sample number in order of increasing Ct values; # indicates conditions where only 1 of 3 replicates amplified, ## indicates conditions where 2 of 3 replicates amplified.
(TIF)

**S6 Fig. Validation of optimal saliva volume.** (A) RT-LAMP results for saliva samples tested on the Bio-Rad CFX96, plotted by Ct value of the corresponding RT-qPCR result, using 5 μL sample volume (left) and 3.75 μL sample volume (right), with a table below summarizing positive percent agreement and 95% confidence intervals at different Ct thresholds. (B) RT-LAMP result of saliva samples on the Axxin T8, plotted by Ct value of the corresponding RT-qPCR result, using 10 μL sample volume (left) and 7.5 μL sample volume (right), with a table below summarizing positive percent agreement and 95% confidence intervals at different Ct thresholds.
(TIF)

**S7 Fig. RT-LAMP performance compared to RT-qPCR on nasal swabs.** (A) Nasal RT-qPCR results are stratified by the Ct value of the paired nasopharyngeal swab. The table below the graph summarizes percent agreement and 95% confidence intervals at different Ct thresholds. (B) Saliva LAMP and Nasal LAMP results from samples tested on the Bio-Rad CFX96 are stratified by the Ct value of the paired nasopharyngeal swab, with tables summarizing percent agreement and 95% confidence intervals at different Ct thresholds.
(TIF)

**S8 Fig. Cumulative presumptive positive samples in LAMP testing.** Cumulative number of samples with presumptive positive results by date. Dashed line: university guidance that fully vaccinated individuals no longer need to test (23 April 2021); dotted line: university guidance that fully vaccinated individuals should resume testing (3 August 2021).
(TIF)

**S1 Table. Test comparison.** Summary of estimated equipment costs, reagent and consumable costs, personnel time, user steps, and limit of detection for RT-qPCR tests and RT-LAMP tests for SARS-CoV-2.
(TIF)

**S1 Data.**
(XLSX)

## Acknowledgments

We gratefully thank Dr. Nathan Butlin and Dr. Nathan Tanner for helpful discussions and thoughtful feedback throughout this project. We also thank the numerous collaborators who helped make the clinical studies possible: Rebecca Elias for assistance with Institutional Review Board protocol submissions; Dr. Abbhi Rajagopal, Mark Munsell, Joseph Thomas, and Jackson Coole for help establishing databases in REDCap; Liam Sweeney, Charles Swope, Kaye Tryels, Sonya Polk-Davis, Angela Muhammad-Ali, Shayla Elliot, Lara Medina, Chelsea Ja'nae, Jessica Thornell, Olalekan Ojeshina, and Nancy Arevalo for their assistance in obtaining samples for initial test development; Arturo Barrera, Jessica Gallegos, Cindy Melendez, Ana López, Keiry Paiz, Jose Garcia, and Anthony Price for clinical coordination at Lyndon B. Johnson Hospital; and Anthony Price and Dr. Ellen Baker for coordinating sample collection during scale-up of the RT-LAMP at Rice University. We gratefully acknowledge Emilie Newsham for help testing clinical samples and Imran Vohra for photography. We thank Dr. Kevin Kirby, Dr. Yousif Shamoo, Jerusha Kasch, Tanner Gardner, Kate Abad, Tina Villard, Lisa Basgall, Abigal Tucker, Leon Lian, and Dr. Jessica McKelvey from Rice University's Crisis Management Advisory Council, Crisis Management Team, and Student Health Centers for support and coordination of the scale-up of the RT-LAMP test at Rice University. Finally, we gratefully acknowledge our student staff members who ran our surveillance testing site, including Norman Zheng, Parker Towns, Keith McCord, Myles Nobles, Grace Waterman, Prince Alino, Keziah Chow, Brittany Bui, George Hung, Emilie Newsham, Angelica Torres, Trinity King, and Lourdes Vivas De Lorenzi.

## Citation diversity statement

Recent work in several scientific fields has identified a bias in citation practice, specifically that publications written by scholars from underrepresented and historically excluded backgrounds and identities are under-cited relative to the number of papers in the field. We recognize this bias and have worked diligently to assess the inclusivity of our reference list and to ensure that we reference appropriate papers from diverse authors.

## Author Contributions

**Conceptualization:** Kathryn A. Kundrod, Mary E. Natoli, Megan M. Chang, Ellen Baker, Kathleen M. Schmeler, Rebecca Richards-Kortum.

**Data curation:** Kathryn A. Kundrod, Mary E. Natoli, Megan M. Chang.

**Formal analysis:** Kathryn A. Kundrod, Mary E. Natoli, Megan M. Chang.

**Funding acquisition:** Kathryn A. Kundrod, Mary E. Natoli, Ellen Baker, Kathleen M. Schmeler, Rebecca Richards-Kortum.

**Investigation:** Kathryn A. Kundrod, Mary E. Natoli, Megan M. Chang, Chelsey A. Smith, Sai Paul, Dereq Ogoe, Christopher Goh, Akshaya Santhanaraj, Anthony Price.

**Methodology:** Kathryn A. Kundrod, Mary E. Natoli, Megan M. Chang, Chelsey A. Smith, Sai Paul, Ellen Baker, Kathleen M. Schmeler, Rebecca Richards-Kortum.

**Project administration:** Kathryn A. Kundrod, Mary E. Natoli, Ellen Baker, Kathleen M. Schmeler, Rebecca Richards-Kortum.

**Resources:** Karen W. Eldin, Keyur P. Patel, Ellen Baker, Kathleen M. Schmeler, Rebecca Richards-Kortum.

**Software:** Dereq Ogoe, Christopher Goh.

**Supervision:** Kathryn A. Kundrod, Mary E. Natoli, Ellen Baker, Kathleen M. Schmeler, Rebecca Richards-Kortum.

**Validation:** Kathryn A. Kundrod, Mary E. Natoli, Megan M. Chang, Chelsey A. Smith, Sai Paul, Dereq Ogoe, Christopher Goh, Akshaya Santhanaraj.

**Visualization:** Kathryn A. Kundrod, Mary E. Natoli, Megan M. Chang, Chelsey A. Smith, Sai Paul.

**Writing – original draft:** Kathryn A. Kundrod, Mary E. Natoli, Megan M. Chang.

**Writing – review & editing:** Kathryn A. Kundrod, Mary E. Natoli, Megan M. Chang, Chelsey A. Smith, Christopher Goh, Karen W. Eldin, Ellen Baker, Kathleen M. Schmeler, Rebecca Richards-Kortum.

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
