## [Decision Letter · Decision Letter 0]

20 Sep 2021

PONE-D-21-25506Sample-to-answer, extraction-free, real-time RT-LAMP test for SARS-CoV-2 in nasopharyngeal, nasal, and saliva samples: Implications and use for surveillance testingPLOS ONE

Dear Dr. Richards-Kortum,

Thank you for submitting your manuscript to PLOS ONE. After careful consideration, we feel that it has merit but does not fully meet PLOS ONE’s publication criteria as it currently stands. Therefore, we invite you to submit a revised version of the manuscript that addresses the points raised during the review process.

ACADEMIC EDITOR: As appended below, the reviewers have raised major concern/critique and suggested further justification/work to consolidate the findings. Do go through the comments and amend the MS accordingly. 

We look forward to receiving your revised manuscript.

Kind regards,

A. M. Abd El-Aty

Academic Editor

PLOS ONE

Journal Requirements:

a. You may seek permission from the original copyright holder of Figure(s) [#] to publish the content specifically under the CC BY 4.0 license.  

Reviewers' comments:

Reviewer's Responses to Questions

**Comments to the Author**

1. Is the manuscript technically sound, and do the data support the conclusions?

Reviewer #1: Partly

Reviewer #2: Yes

Reviewer #3: Yes

2. Has the statistical analysis been performed appropriately and rigorously? 

Reviewer #1: Yes

Reviewer #2: Yes

Reviewer #3: Yes

3. Have the authors made all data underlying the findings in their manuscript fully available?

Reviewer #1: Yes

Reviewer #2: Yes

Reviewer #3: Yes

4. Is the manuscript presented in an intelligible fashion and written in standard English?

Reviewer #1: Yes

Reviewer #2: Yes

Reviewer #3: Yes

5. Review Comments to the Author

Reviewer #1: Overall the manuscript is well written and clear. There are a few sections where some of the methods for comparison are a bit low on numbers or used non-optimal specimens or controls. This does not add any unjustified conclusions, but could have more optimal conditions set. Here are following comments and recommendations for the manuscript.

Major Comments

What is the extraction method for the taqpath assay that was used for the RT-PCR reference method? This assay can have a variety of extraction methods that can have large differences on sensitivity.

What was the rationale for placing NP into 300uls and Nasal into 500uls. Although small, this dilution factor could cause some of the differences between sensitivity when comparing results.

The addition of multiple freeze thaws an cause degradation of specimens. Generally, a specimens should not go through more than 1 freeze thaw. Was any testing performed to determine that the CT values did not increase after each freeze thaw.

What was the rational for testing surveillance specimens in triplicate? Was this to increase sensitivity due to issues found in previous experiments?

Testing a subset of specimens for internal controls is not a traditional method. What was the number of specimens that failed and how often was re-testing needed.

For the surveillance testing, was all specimens also tested on a RT-PCR as a reference method. If so this comparison should be highlighted to discuss PPA and NPA.

Minor Comments

Ln89-103 This section is written as an abstract with data instead of a lead in for the introduction.

Overall the manuscript is a bit long and could have portions removed to make it a more concise manuscript. I would suggest modifying to reduce the length. Some examples that could be removed are parts of the materials and methods for example the NP collection method, methods for reducing cross contamination.

LoD study: The testing that was performed was with a construct quality control material instead of an inactivated virus. Was any testing performed with virus for sensitivity as the construct vs real virus could have downstream effects of extraction methods.

Figure 3 – What was the # of copies per… ml or ul?

When looking at the amount of saliva added to the specimens for saliva, what was the starting CT for those specimens, this can be from the NP sample.

Reviewer #2: A well written manuscript with very detail description of the methods and procedures.

The use of multiple primer sets can increase sensitivity, but it increases primer cost. Need some information on the cost difference.

The authors may want to address cross-contamination issue when the sample preparation step is done manually and pipetted into tubes.

The manuscript should address the total cost (including labor) needed to perform each LAMP test in the near POC (using Axxin device) and high throughput (using the Bio-Rad thermal cycle) settings versus real-time RT-PCR tests performed primarily in high throughput lab.

This work confirms the previous findings that samples with low Ct values were more commonly found in the asymptomatic surveillance population than the hospitalized population.

The manuscript stated that “sample collection, inactivation, and amplification requires <5 user steps, and results can be obtained in <1 hour.” It is unclear if this is for a single sample or the same turn-around-time can be obtained if 8 samples are collected and processed for the Axxin incubator/detector.

Overall, this manuscript provides readers with information on the success of using LAMP assay to perform surveillance testing in a university campus setting. I’d recommend it for publication with minor revision.

Reviewer #3: Thank you for inviting me to peer review the manuscript titled “Sample-to-answer, extraction-free, real-time RT-LAMP test for SARS-CoV-2 in nasopharyngeal, nasal, and saliva samples: Implications and use for surveillance testing" by Kathryn A. Kundrod et al. The authors optimized a RT-LAMP based nucleic acid test for SARS-CoV-2 detection for nasopharyngeal swabs, nasal swabs, and saliva collected from hospitalized patients. They used Hamilton Microlab Prep device to automate the procedure (It should be noted that currently a FDA EUA approved LAMP-based test is also available for home testing, Lucira COVID-19 All-In-One Test Kit, prescribed to be used in PoC and also at a home setting for suspected people older than 14. This is not mentioned in the manuscript). The manuscript is well written and could be interesting to the readers. After careful reading, I decide to accept this paper for publication.

Reviewer # 4:In this study, Kundrod et al. undertake broad validation of RT-LAMP for community-based screening that addresses many fundamental concerns about faster, simpler techniques being developed around the world in the wake of COVID-19. Through laboratory-based comparisons with the clinical standard, detailed description of LOD in different sample types and under various conditions in a clinical cohort, and application in a community-based setting with low incidence, the authors clearly demonstrate the utility of RT-LAMP to detect those with high levels of viral RNA more quickly than the current standard. The study is very well-written, with clearly explained objectives that seem highly relevant to those (like myself) who are new to the field. As a result, I can find very little to criticize about it, and can’t wait to apply the lessons described by the authors in my own work. If possible, it would be nice to see some additional discussion about the cost vs. benefit of the surveillance activities undertaken - how much did each of those 9 cases cost? How low does the cost have to be to justify detection of such a small proportion of positives? Practically speaking, won’t people just stick with symptom screening and hope for the best, because it is exponentially cheaper? The point about repeat testing vs. sensitivity is well-taken, since someone could have a low viral load just prior to peak infectivity. But information about how many asymptomatic individuals were tested how many times is not given. Individuals were tested weekly or bi-weekly, but based on your experience what would be the ideal time interval between samples, to minimize the amount of samples required to monitor a given population? Just out of curiosity, not because I think the paper requires any major revision.

6. PLOS authors have the option to publish the peer review history of their article (what does this mean?). If published, this will include your full peer review and any attached files.

Reviewer #1: No

Reviewer #2: No

Reviewer #3: No

---

## [Author Response · Author response to Decision Letter 0]

3 Nov 2021

Thank you for the opportunity to revise our manuscript. Our response to specific reviewer comments was uploaded as an attached Reviewer Response file.

---

## [Decision Letter · Decision Letter 1]

23 Nov 2021

PONE-D-21-25506R1Sample-to-answer, extraction-free, real-time RT-LAMP test for SARS-CoV-2 in nasopharyngeal, nasal, and saliva samples: Implications and use for surveillance testingPLOS ONE

Dear Dr. Richards-Kortum,

Thank you for submitting your manuscript to PLOS ONE. After careful consideration, we feel that it has merit but does not fully meet PLOS ONE’s publication criteria as it currently stands. Therefore, we invite you to submit a revised version of the manuscript that addresses the points raised during the review process.

ACADEMIC EDITOR: Still, reviewer # 2 is raising a major concern over the revised form of the MS. Do go through the comments and amend the MS accordingly. 

We look forward to receiving your revised manuscript.

Kind regards,

A. M. Abd El-Aty

Academic Editor

PLOS ONE

Reviewers' comments:

Reviewer's Responses to Questions

**Comments to the Author**

1. If the authors have adequately addressed your comments raised in a previous round of review and you feel that this manuscript is now acceptable for publication, you may indicate that here to bypass the “Comments to the Author” section, enter your conflict of interest statement in the “Confidential to Editor” section, and submit your "Accept" recommendation.

Reviewer #1: All comments have been addressed

Reviewer #2: (No Response)

2. Is the manuscript technically sound, and do the data support the conclusions?

Reviewer #1: Yes

Reviewer #2: Yes

3. Has the statistical analysis been performed appropriately and rigorously? 

Reviewer #1: Yes

Reviewer #2: Yes

4. Have the authors made all data underlying the findings in their manuscript fully available?

Reviewer #1: Yes

Reviewer #2: Yes

5. Is the manuscript presented in an intelligible fashion and written in standard English?

Reviewer #1: Yes

Reviewer #2: Yes

6. Review Comments to the Author

Reviewer #1: The manuscript submitted by Richards-Kortum et al entitled “Samples-to-answer, extraction-free real-time RT-LAMP test for SARS-CoV-2 in NP, NS, and saliva samples: Implications for surveillance testing” describes a study where the authors developed a RT-LAMP assay for detection of SARS-CoV-2 from NP, NS, and saliva samples. The assay uses 3 viral targets and an internal human control. Initially they optimized the assay for extraction and amplification using synthetic controls. When optimized they validated the assay with previous positive hospitalized patients. Results from these studies are similar with others where NP was most sensitive followed by NS and saliva samples. Also similar to other publications isothermal LAMP showed a reduced sensitivity compared to traditional RT-PCR assays. The authors then used the assay for a local screening of 400 individuals. A subset of positives was confirmed positive by a RT-PCR instrument. Overall, the manuscript is well written; however, these data are similar to other assays on the market. The most novel aspect is the surveillance testing, which could use some additional details and clarity. Prior to submission the following comments and suggestions should be addressed.

Major Comments

Validation patient samples were all identified as previous positive patients which can skew results for the validation due to a higher-than-average prevalence. I would suggest increasing the number of negative samples tested to adequately evaluate the specificity of the assay or at minimum discuss this as a limitation.

Figure 7: In the abstract the follow up screening was performed using 400 individuals, but the numbers in the text ln668 had 20,645 and all of these seem to be nasal and not saliva. The figure also has these listed out as nasal swabs. Additional clarification is needed for these numbers. Also, why were only 8 of the 136 followed up with confirmatory RT-qPCR testing. Could there be a risk of FP results as the NPA was not adequately evaluated in the validation.

You mention that during the screening period you saw an increase in positives with emergence of Delta variant. What percentage of the screened specimens were found to be presumptive positive during the Delta surge? This may be a reason why you see lower CT values in this cohort compared to hospitalized patients as Delta appears to have a higher viral load.

The manuscript is very detailed, but at this point of the pandemic there are several manuscripts detailing results that were found in the optimization of the assay and the most novel aspect of surveliiance testing at Rice is quickly summarized in the results and discussion. My suggestion is to move most of the optimization data to supplemental and create a short summary few paragraphs in either the results or discussion. This would allow you to expand discussion for the validation and surveillance data

Minor Comments

Abstract: Add data for > Ct 30

Introduction: Although it is well written, the introduction could be cut down. With the amount of literature in the field on SARS-CoV-2 I would remove lines 58-66 and could likely combine the first paragraph and the third.

Ln301: is the “was not” a typo, or did you use assent template for minors?

As you went back to patients that were positive for the validation, how long was the new collection from time of collection, especially as you continually discuss the viral load vs time course of infection.

Reviewer #2: (No Response)

7. PLOS authors have the option to publish the peer review history of their article (what does this mean?). If published, this will include your full peer review and any attached files.

Reviewer #1: No

Reviewer #2: No

---

## [Author Response · Author response to Decision Letter 1]

23 Dec 2021

Thank you for the opportunity to revise our manuscript. Our responses to the reviewer comments are included in the uploaded Response to Reviewers document.

---

## [Decision Letter · Decision Letter 2]

4 Feb 2022

Sample-to-answer, extraction-free, real-time RT-LAMP test for SARS-CoV-2 in nasopharyngeal, nasal, and saliva samples: Implications and use for surveillance testing

PONE-D-21-25506R2

Dear Dr. Richards-Kortum,

We’re pleased to inform you that your manuscript has been judged scientifically suitable for publication and will be formally accepted for publication once it meets all outstanding technical requirements.

Kind regards,

A. M. Abd El-Aty

Academic Editor

PLOS ONE

Additional Editor Comments (optional):

Reviewers' comments:

Reviewer's Responses to Questions

**Comments to the Author**

1. If the authors have adequately addressed your comments raised in a previous round of review and you feel that this manuscript is now acceptable for publication, you may indicate that here to bypass the “Comments to the Author” section, enter your conflict of interest statement in the “Confidential to Editor” section, and submit your "Accept" recommendation.

Reviewer #4: (No Response)

2. Is the manuscript technically sound, and do the data support the conclusions?

Reviewer #4: Yes

3. Has the statistical analysis been performed appropriately and rigorously? 

Reviewer #4: Yes

4. Have the authors made all data underlying the findings in their manuscript fully available?

Reviewer #4: No

5. Is the manuscript presented in an intelligible fashion and written in standard English?

Reviewer #4: Yes

6. Review Comments to the Author

Reviewer #4: I have reviewed the paper in its current form, and as a whole the work seems to have been performed in a reasonable manner. I have two observations:

1. I do not agree with the author's decision to ignore a reviewer's suggestion that optimization information be placed in a supplemental appendix. I think the manuscript would be more readable if this were done. I suspect the vast majority of readers will just skim or skip over this part. My objection is not sufficiently strenuous to make me wish to bar publication on this basis, but I believe the editors should give this consideration.

2. The authors state that all their data is available, but they do not provide a link to a repository for deidentified data, or a supplemental appendix with full data. Thus, I DO NOT believe they have met the requirement for data availability.

Possibly I missed something in my review, but I did not find this information in either the original submission or either revision.

7. PLOS authors have the option to publish the peer review history of their article (what does this mean?). If published, this will include your full peer review and any attached files.

Reviewer #4: No

---

## [Editor Report · Acceptance letter]

16 Feb 2022

PONE-D-21-25506R2 

Sample-to-answer, extraction-free, real-time RT-LAMP test for SARS-CoV-2 in nasopharyngeal, nasal, and saliva samples: Implications and use for surveillance testing 

Dear Dr. Richards-Kortum:

I'm pleased to inform you that your manuscript has been deemed suitable for publication in PLOS ONE. Congratulations! Your manuscript is now with our production department. 

Kind regards, 

on behalf of

Prof. A. M. Abd El-Aty 

Academic Editor

PLOS ONE